# A Review on Reliability of Integrated Electricity-Gas System

You Zhou and Chuan He *

College of Electrical Engineering, Sichuan University, Chengdu 610065, China
* Correspondence: he_chuan@scu.edu.cn

**Abstract:** With the development of society and the increase in energy demand, the electric power system and natural gas system become larger and larger, where the network is increasingly complex. The continuous and stable supply of energy is also important. The reliability evaluation of the power system and natural gas system is an essential part of maintaining efficient operation of the system. In this paper, the concept of the integrated electricity-gas system is introduced first, and then the importance of the reliability evaluation of the power system and natural gas system is emphasized. In addition, the main reliability indices and reliability evaluation method of the two systems are presented, and some practical examples are given. Finally, conclusions on the reliability evaluation of the integrated electricity-gas system are drawn.

**Keywords:** reliability indices; power system; evaluation methods; natural gas system; integrated electricity-gas system

## 1. Introduction

With the modernization and informationization of society, demand for power supply is increasing. A high-quality, reliable, and stable power supply is the basis for promoting social development, scientific and technological innovation, and national economic strength. According to the China Energy Data Report (2021), China's total power generation reached 7.78 billion kWh in 2020, which is an increase of about 85 percent over the whole year of 2010. Based on the energy crisis and environmental degradation, all countries are striving to adjust their energy structure, diversify their energy sources, develop renewable energy, and complete electricity reform. To this end, the United States has put forward the strategic requirements for the smart grid to build a secure, reliable, and flexible integrated energy network [1]. Additionally, China has proposed the construction of the energy internet [2], while European countries have also carried out in-depth research and practice on integrated energy systems [3]. The proposal of the concept of the integrated energy system provides a reference direction for the development of the power system [4,5]. The integrated energy system represented by the integrated electricity-gas system (IEGS) has been operated in Denmark, the United States, China, and other countries. The IEGS plays an important role in the accommodation of wind, solar, and other renewable energy sources and the reduction of carbon emissions. The natural gas system can store excessive renewable energy of electric power via power-to-gas (P2G) and supply it to the natural gas loads or generate electricity through gas-fired units.

While IEGS brings benefits such as energy-efficient utilization, new operational risks are also brought to the natural gas system and power system. At the same time, more stringent requirements have also been put forward by the society for power supply, not only for the economics of power generation, but also for the security and reliability of the power supply. In the daily operation of the IEGS, it is inevitable that some major natural disasters, equipment quality problems, equipment aging, misoperation, or other factors lead to abnormal operating states of the power system, which results in occurrence of power outages. For instance, Texas was affected by extreme cold weather in 2021, where the gas production from natural gas wells was restricted, and pipeline failures resulted

in the forced outage of gas-generating units. In order to ensure the energy balance of the power grid, the load had to be curtailed, which eventually led to power outages for about 5 million customers [6]. Coincidentally, in 2017, the gas-fired units of Taiwan's Tai Tan power plant in China were shut down due to gas supply disruptions, resulting in widespread power outages [7]. In this power outage, about 60% of users were affected by power outages [8]. Furthermore, in 2019, the Hornsea wind farm was taken off the grid due to oscillation of the renewable energy power plant, which led to a series of chain reactions that caused the frequency of the grid to drop. This eventually led to power outages in some cities in the UK, and about 1 million people were affected [9,10]. Once these power outages happen, they cause inconvenience in daily life, bring about huge economic loss to society, and even endanger personal security. So, the reliability assessment of the power grid and gas network is extremely important. Some specialized institutions have been established to evaluate the power system reliability to reduce the social and economic impact of such blackouts, such as the Power Reliability Management Center of the China Electricity Council, the North American Electric Reliability Corporation [11], and other institutions. This shows the importance of evaluating the reliability of the power grid.

The reliability assessment of the power system, natural gas system, and IEGS is discussed in this paper. Therefore, the authors have discussed reliability evaluation metrics for these systems in Section 2. Then in Section 3, some existing reliability assessment methods are introduced, and related applications are given. Finally, the author gives an outlook on the reliability assessment of IEGS. The outline of the full paper is shown in Figure 1.

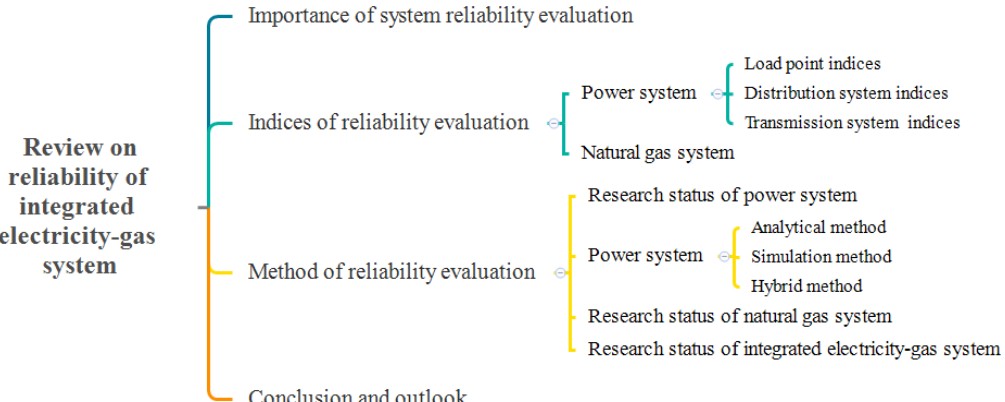

**Figure 1.** Outline of the paper.

## 2. Reliability Indices

Reliability is defined as a measure of the ability of a device or electrical system or natural gas system to perform its intended function under specified operating conditions and within a set time frame [12]. The intended function is usually the ability of the power system or natural gas system to provide users with uninterrupted high-quality power [13] or natural gas. The quantitative evaluation standard of power system reliability is defined based on the number of power outages, the time of power supply restoration, and the degree of power outage influence [14]. Usually, power system reliability evaluation can be divided into three types: (1) reliability assessment of the power generation and transmission system; (2) reliability assessment of power plants and substations; (3) reliability assessment of the power distribution system. Researchers have formerly paid more attention to the reliability of the integrated generation and transmission system. When the power supply is insufficient, the stable supply of power to the power generation and transmission system is particularly important.

Compared with generation and transmission systems, the reliability of distribution systems has not received much attention in the initial stage, as the failure of distribution networks often occurs only in a certain area that supplies few customers, where the impact

of the fault is smaller. Therefore, the research on distribution system reliability starts late. Now, with the increase of the power load, the distribution network is more and more complex. As the last mile of the power supply in the power system, the distribution system is directly connected with the customers. At the same time, it also reflects the quality and ability of the power supply. Because the distribution network is generally radial, if a component or line fails, it will affect the users downstream. Therefore, it is very necessary to ensure the reliable operation of the power system, which has also become routine work in the operation and planning of the power system. There are many methods to improve the reliability of the distribution network, such as the optimal planning of protective devices, which can improve the reliability of distribution network operation. A mixed-integer nonlinear programming model is proposed in [15] to determine the type, installation location, and quantity of protective devices such as segment switches and fuses considering reliability-related costs. In addition, microgrids could operate in islanding mode, and could also improve the reliability of distribution networks. In reference [16], a loop-based topology design constraint for microgrid topology planning is proposed, and the optimal loop is determined by gradually discarding the infeasible and non-optimal structure in the active distribution network. Responsive load is also an effective way to improve the reliability of the power system [17]. The strengthening of transmission lines can also improve the operational reliability of the transmission network. In reference [18], based on the two extreme weather conditions of strong wind and strong lightning, the function of reliability with respect to weather conditions is established, and the failure rate and recovery time of overhead lines are considered in the model. The reliability evaluation indices and methods of the power system and natural gas system are introduced in this paper. At present, the well-known and widely accepted reliability evaluation standards mainly include MIL-HDBK-217, Telcordia SR-332, 217Plus, NSWC Mechanical, ANSI/VITA 51.1, China's GJB/z 299 standards, and NPRD and EPRD. Typically, the transmission line failure rate is 0.01–0.1 frequency (per km-year), the transformer failure rate is about 0.015 frequency (per year), and the generator is about 0.03 frequency (per year) [19]. From the US Department of Transportation databases in the most recent years, the gas pipeline failure rate is relatively close at $8.9 \times 10^{-5}$ frequency (per km-year) [20]. Table 1 provides a summary of the reliability indices presented in this paper.

**Table 1.** Summary of reliability indices in the paper.

| Index Properties | | Index Name | Unit |
|---|---|---|---|
| Power system | Load Point Indices | The average annual outage time of load point | h/year |
| | | The annual outage rate at load point | times/year |
| | | The average failure repair time of load point | h/time |
| | | Energy not supplied | MWh/year |
| | | Annual power supply reliability of load point | - |
| | Distribution System Reliability Indices | System average interruption frequency | Times (customer year) |
| | | System average interruption duration | Hours (customer year) |
| | | Customer average interruption duration | Hours (customer time) |
| | | Average service availability | - |
| | | Average energy not supplied | MWh/customer |
| | | The total equivalent outages times | h |
| | Transmission System Reliability Indices | Loss of load probability | - |
| | | Loss of load expectation | h |
| | | Expectation energy not supplied | MWh |
| | | Loss of load frequency | times/year |
| | | Mean time between failures | h |
| Natural gas system | | Failure rate | - |
| | | Reliability | - |
| | | Availability | - |

*2.1. Power System Reliability Indices*

Generally, power system reliability indices can be divided into two categories: load point indices and system indices. The first one is mainly for customers to represent the local impact of the fault. The other is for the system to measure the impact of the failure on the entire system from a global perspective.

### 2.1.1. Load Point Indices

From the perspective of power system users, the reliability indices of the load point quantitatively evaluate the degree of continuous and uninterrupted power consumption by power users. The reliability of the power supply at a load point is evaluated by the average annual outage time of the load point, the annual outage rate at the load point, the average failure repair time of the load point, the energy not supplied index, and the annual power supply reliability rate. These five indices are briefly described below.

(1)  The average annual outage time of the load point, (hours per year). This index is mainly used to reflect the average outage time experienced by power customers each year.

(2)  The annual outage rate at the load point, (times per year). The index measures the average number of blackouts that each user experiences each year over a statistical time frame, including both short-term blackouts and persistent blackouts. Short-term power outages are defined as power outages of less than three minutes according to the national standard.

(3)  The average failure repair time of load point, $\gamma_i^e$ (hours per time). This index measures the average time between each blackout and the restoration of the power supply. It also reflects the power system staff for the emergency power outage handling efficiency. Usually, this value is relatively small in the case of operations with backup components or other power supply methods. The relation of the above three load point reliability indices can be expressed by the following equation.

$$\gamma_i^e = \frac{U_i}{\lambda_i^e} \tag{1}$$

where $U_i$ represents the average annual outage time at load point $i$, $\lambda_i^e$ represents the annual outage rate at load point $i$.

(4)  Energy not supplied, *ENS* (MWh per year). This index calculates the expected value of the power shortage every year at the load point to measure the severity of the power shortage for the user. The calculation is as follows:

$$ENS_i = L_i \cdot U_i \tag{2}$$

where $L_i$ represents the average load at load point $i$.

(5)  Annual power supply reliability of load point, $R_i$. This index calculates the proportion of the time for obtaining the power supply in one year at the load point, which directly reflects the ability of the power system to supply power to the users. It is calculated as follows:

$$R_i = (1 - \frac{x_i}{T}) \times 100\% \tag{3}$$

where $x_i$ is the total duration of the power outage in a year at load point $i$, $T$ represents 8760 h in a year.

### 2.1.2. Distribution System Reliability Indices

The reliability indices of the load point can only reflect the reliability of the power system from the customer side, while the reliability index of the power system can evaluate the overall power supply capability of the power system. The purpose of the power system reliability assessment is to characterize the degree of operational reliability. The reliability

indices of the power system are usually divided into the following three categories: (1) the indices defined by the probability of failure, such as reliability, availability, and average number of failures; (2) the indices defined by the time of the fault, such as the average power outage time, the average time to repair the fault, the expected days of failure, etc.; (3) the indices defined by the extent of the failure, such as the average reduction of power supply expectations. The following are some of the more widely used reliability indices [21–23].

(1) System average interruption frequency index, $SAIFI$ (times per customer per year). This index calculates the average number of outages experienced by each customer per unit time in the distribution network, which is shown as follows. According to IEEE Standard 1366–1998, the median value for North American utilities is approximately 1.10 interruptions per customer.

$$SAIFI = \frac{\sum\limits_{j \in \rho} N_j^{\mathrm{e}}}{N^{\mathrm{e}}} \tag{4}$$

where $N_j^{\mathrm{e}}$ is the number of interrupted customers in outage event $j$; $N^{\mathrm{e}}$ is the total number of customers served in the area; $\rho$ represents the set of all outage events in the distribution network. Obviously, lower values of SAIFI represent higher reliability of the distribution system. If we want to improve the reliability of the power system, we need to reduce the occurrence of power outages and the failure rate of customers.

(2) System average interruption duration index, $SAIDI$ (hours per customer per year). SAIDI is determined by dividing the sum of all customer interruption durations during a year by the number of customers served. This indicator calculates the duration of the customer's average power outage per unit time, which is usually one year, and its calculation is as follows. According to IEEE Standard 1366–1998, the median value for North American utilities is approximately 1.50 hours.

$$SAIDI = \frac{\sum\limits_{j \in \rho} U_j N_j^{\mathrm{e}}}{N^{\mathrm{e}}} \tag{5}$$

where $U_j$ represents the average annual power outage time of outage event $j$.

(3) Customer average interruption duration index, $CAIDI$ (hours per customer per time). This index is determined by dividing the sum of all customer interruption durations by the number of customers experiencing one or more interruptions over a one-year period, and is calculated as follows.

$$CAIDI = \frac{\sum\limits_{j \in \rho} U_j N_j^{\mathrm{e}}}{\sum\limits_{j \in \rho} N_j^{\mathrm{e}}} \tag{6}$$

As can be seen from the equation, CAIDI can also be obtained by dividing the value of SAIDI by the value of SAIFI. All three indicators are for consumers, who can be individuals, factories, or companies that use electricity.

(4) Average service availability index, $ASAI$. This index gives the fraction of time in which the customer has power during the reporting time. The calculation of this indicator is as follows.

$$ASAI = \frac{\sum\limits_{j \in \rho} 8760 N_j^{\mathrm{e}} - \sum\limits_{j \in \rho} U_j N_j^{\mathrm{e}}}{\sum\limits_{j \in \rho} 8760 N_j^{\mathrm{e}}} = \frac{8760 - SAIDI}{8760} \tag{7}$$

The value of the index is proportional to the reliability of the distribution system, and the algebraic relation between the index and SAIDI is given in the equation.

(5)  Average energy not supplied, *AENS* (MWh per customer). This index calculates the average power shortage of customers in power outage accidents caused by the failure of the generator and other power-generating equipment in a year. The calculation for this index is as follows.

$$AENS = \frac{\sum\limits_i P_i^{avg} U_i}{\sum\limits_i N_i^e} \tag{8}$$

$$P_i^{avg} = \frac{\sum\limits_t P_{it}}{T} \tag{9}$$

where $P_i^{avg}$ represents the average load at load point $i$, $N_i^e$ is the number of customers at load point $i$, $P_{it}$ represents the electricity demand at time $t$.

2.1.3. Transmission System Reliability Indices

Different from the distribution network, the transmission network plays the role of a bridge from the power plant to the customers. However, the reliable operation of the transmission grid is an important factor affecting the stable power supply of the power system. The Electric Power Research Institute, in cooperation with many parties, proposes a set of reliability evaluation indicators for the transmission network. For long-distance, high-power DC electric power transmission, the reliability index is defined from the perspective of transmission capacity. Here are some commonly used main power grid reliability assessment indicators:

(1)  The total equivalent outages time, *TEOT* (hours), which calculates the duration of each downtime in a transmission system, is converted into the equivalent outages of the rated transmission capacity by the ratio of the downtime to the rated transmission capacity. The expression is as follows:

$$TEOT = \sum_l \frac{P_l^0}{P^m} \times t_l \tag{10}$$

where $l$ represents the index of the system in the reduced operating state, $P_l^0$ is the outage capacity when the system is in the derated operation, $P^m$ is the rated transmission capacity of the system, $t_l$ represents the duration in which the system is in a reduced operating state.

(2)  Loss of load probability, *LOLP*. This index calculates the probability of power outages occurring when the total power generation capacity of the system is less than the load demand during unit time, or when the power outage occurs because of the occurrence of faults. Its calculation formula is as follows:

$$LOLP = \sum_{j \in F} p_j \tag{11}$$

where *F* is the set of outage accidents of power generation equipment in a certain time, $p_i$ is the probability of outage event $j$ occurring in the transmission system.

(3)  Loss of load expectation, *LOLE* (hours). This index calculates the expected number of days in which the system will experience load shedding in a unit of time (usually 1 year). For example, the French LOLE indicator standard is an average of 3 h per year. This number is based on a forecast of 30 h of disruption per decade. According to the European Commission Decision on Polish capacity mechanism, the reliability standard set for the Polish electricity market is equal to a LOLE of 3 h per annum [24]. According to the MISO reliability standard, LOLE should meet less than 0.1 days per year [25]. This indicator is a derived indicator of LOLP, and its calculation is as follows:

$$LOLE = T \times LOLP \tag{12}$$

(4) Expectation energy not supplied, *EENS* (MWh). This indicator calculates the total amount of power generation that is reduced by the system due to outages of power generation equipment or other failures on an average yearly basis. At the same time, this indicator also represents the average power value that the system lacks in operation each year. The calculation is as follows:

$$EENS = T \times \sum_{j \in F} p_j c_j \tag{13}$$

where $c_j$ is the power shortage in each outage event.

(5) Loss of load frequency, *LOLF* (times per year). This index calculates the number of power outages that occur in the system per year. The formulation is as follows:

$$LOLF = \sum_{j \in F} f_j \tag{14}$$

where $f_j$ represents the transition frequency at which the system reaches the normal operating state after only one state transition.

According to [26], although some reliability indices can effectively evaluate the probabilistic reliability of the power system, they usually cannot reflect the influence of some original systems and subsystems on the overall system reliability; it is also not possible to accurately model the failure model of each component separately. Therefore, a Bayesian-network-model-based reliability assessment of power systems is proposed, and the accuracy of LOLP is the same as other methods, as is verified. At the same time, this method is more conducive to finding out the reasons as to why the reliability of the power system is affected, as well as the weak links in the power grid. Similarly, in [27], the authors propose a new metric to measure reliability for users' instantaneous interruptions and storms. The MAIFI proposed by IEEE is used to replace SAIFI to evaluate the impact of instantaneous interruption on system reliability, and the Storm Average Interruption Duration Index (STATDI) is proposed to replace SAIDI to evaluate the impact of storm events on system reliability.

### 2.2. Natural Gas System Reliability Indices

The reliability of the natural gas system aims at two aspects. The first aspect is operation reliability. The second aspect is reliable gas supply. The natural gas system has sufficient gas sources and sufficient pipeline transmission capacity, which can ensure the daily use of users and the natural gas demand during peak hours. In the operation reliability of the natural gas system, there are indices such as mean time between failures, mean fault repair time, failure rate, repair rate, reliability, availability, and so on. The following is a description of several commonly used reliability indicators of natural gas systems.

(1) Mean time between failures, MTBF (hours). This index represents the average continuous running time of each piece of equipment in the natural gas system after each maintenance period or after it is newly put into use to the next failure.

$$T_f = \frac{1}{\sum\limits_{m \in N^g} n_m} \sum_{m \in N^g} \sum_{k \in n_m} t_{mk} \tag{15}$$

where $N^g$ is the total number of pipelines in the natural gas system, $m$ is the index of the pipeline, $n_m$ represents the number of failures of the $mth$ pipeline per unit time, $t_{mk}$ represents the working time of the pipeline from the last failure to the $kth$ failure.

(2) Failure rate, $\lambda^g$. This indicator represents the probability of failure of the equipment in the natural gas system within a unit time, excluding the maintenance time, and the unit is Fit (1Fit means that only one failure occurs within $10^9$ h). If the service

life of the equipment follows an exponential distribution (which is satisfied by most pipelines and equipment), the algebraic relationship between this index and MTBF is as follows:

$$\lambda^g = \frac{1}{T_f} \tag{16}$$

(3) Reliability, *R*. This index indicates the probability that the research equipment in the natural gas system will not fail in the specified time t under the specified operating conditions. According to the *Fault Tree Handbook* NUREG-0492 [28], the distribution of faults obeys the exponential law under some assumptions. (a). When the equipment has just been put into operation or has been recommissioned after a repair $R(t) = 1$. (b). The reliability $R(t)$ is a single-valued decreasing function at time $t$. Different time $t$ corresponds to different reliability, when $t \to \infty$, $R(t) \to 0$. (c). For the process of equipment use ($0 < t < \infty$), there is always $0 < R(t) < 1$. The reliability and failure rate of most devices satisfy the following relations:

$$R(t) = e^{-\lambda^g t} \tag{17}$$

(4) Availability, *A*. This index represents the probability that the maintainable equipment can work normally until a certain time t under specified conditions. For equipment such as pipelines in a natural gas system, maintenance can continue to function normally as the run time increases. In this case, the failure rate does not reflect the reliability of operation well. If the life of the equipment follows an exponential distribution, the availability is related to the failure rate as follows:

$$A = \frac{\mu^g}{\mu^g + \lambda^g} \tag{18}$$

where $\mu^g$ is the maintenance rate of the equipment, representing the average maintenance times of the equipment in unit time, which is also reciprocal to the average maintenance time of the equipment, $\lambda^g$ is the failure rate.

In addition to these indices, there are some reliability indices similar to the power system, which extend the concept of the power system reliability index to the natural gas system. From this, indicators such as the average gas shortage, the system average interruption frequency index (SAIFI), and the expected gas not supplied index (EGNS) are defined.

## 3. Reliability Evaluation Method

### 3.1. Research Status of Power System Reliability Evaluation Methods

The traditional power system reliability assessment is based on the reliability model of each component in the system to calculate the reliability of the load point. According to the definition of the reliability index, each reliability value of the whole power system is obtained. There are mainly three methods for reliability evaluation of power systems: analytical method, simulation method, and hybrid method. The first two methods are similar as they both need to first calculate the probabilities of different states of components in the system, and then calculate and analyze the consequences. The analytical method is to establish the reliability model of all components in the system, enumerate various possible fault conditions, and use the numerical calculation method to obtain the probability of occurrence of the corresponding state. Different from this, the simulation method obtains the possible states of the system through a large number of random sampling, and obtains the probability of occurrence of different states of the system through statistical methods. Among the two methods, the analytical method is faster and more accurate, but the calculation process is more complicated and this method is not suitable for processing larger systems. The simulation method is more suitable to deal with complex and huge systems, but the calculation precision of this method is lower. The hybrid method is a combination of these two methods, taking the advantages of the first two methods.

However, compared with the first two methods, there are fewer studies on the hybrid method, and further exploration is needed. The main reliability evaluation methods are classified as presented in Figure 2.

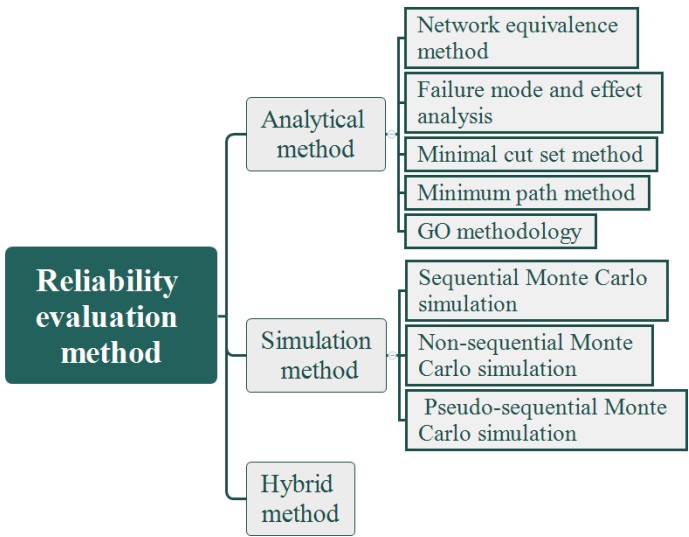

**Figure 2.** Classification of evaluation methods.

A comprehensive reliability assessment method is proposed in [29], which can be used to assess the reliability of power generation systems, transmission systems, and distribution systems. A distribution network expansion planning model that takes the reliability index into account is presented in [30], which considers fault recovery between lines and redistribution of power flow. A method of hierarchical Markov modeling is proposed in [31] and is used to analyze the reliability of power distribution systems. The method decomposes the system topology and obtains a reliability model considering the protection device, which is convenient for analyzing multiple faults. A method with cost and the reliability of distribution systems as objective functions is proposed in [32] to locate potential feeder additions in islanded distribution networks with distributed renewable energy sources and energy storage devices. A simulation-based distribution network planning model with a reliability index as the basis for scheme selection is proposed in [33]. The concept of decision-dependent uncertainty is introduced in the power system operational reliability assessment in [34], and is considered in a two-stage stochastic unit commitment formulation to quantify reliability metrics. In [35], the authors propose a method that can be used to assess the reliability of complex radial distribution networks containing devices such as circuit breakers, transformers, manual disconnectors, and automatic disconnectors. The article shows that more automatic switches in operation can improve the reliability of the system. In [36], the authors use the Monte Carlo simulation method to evaluate the reliability of the active distribution system with renewable distributed generators. In order to improve the solution efficiency of the model, techniques such as two-step sampling, region division, and minimal path search are also used to speed up the solution process. The reliability evaluation model of the distribution system including the access of electric vehicles is proposed in [37], which shows that the integration of electric vehicles can contribute to the improvement of distribution system reliability. In reference [38], a method for calculating the reliability index of the distribution network is proposed, which uses analytical methods and Monte Carlo simulation methods to simulate the fault state of components. Two methods for reliability assessment of power distribution systems are introduced in [39], namely historical assessment and predictive assessment. In [40], two parallel sequential Monte Carlo simulation methods are proposed for reliability assessment of power systems. In [41], the authors first search for the most probable states that lead to power system failures based on the genetic algorithm, and then analyze the annualized reliability indices.

### 3.2. Power System Reliability Evaluation Methods

3.2.1. Analytical Method

Desieno and Stine extended the mathematical model of Markov process to power system reliability assessment in 1964 [42], where an analytical method was established. The analytical method focuses on considering the logical relationship between the components and the network topology in the system. Based on the random parameters of each element in the system, the reliability model of the system is established by the analytical method. Then, one must enumerate all possible fault conditions, analyze the components under each fault condition, and calculate each reliability index. The analytical method is mainly used for systems where the topology is relatively simple and small-scale [43]. However, when analyzing large-scale systems, the computational complexity of this method increases exponentially with the number of system components. A brief flow chart of the method is presented in Figure 3.

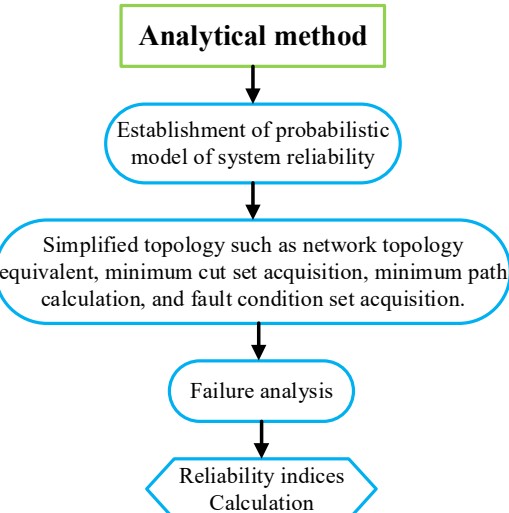

**Figure 3.** Flow chart of the analytical method.

(1)    Network Equivalence Method

Based on the principle of network equivalence simplification, the network equivalence method simplifies the complex network topology with branch structure equivalent to the relatively simple radial network topology. Next, the reliability index of the whole system is calculated according to the equivalent conversion formulation of the reliability index. In the process of equivalence, the component is used to replace part of the network, so the complex relationship between the component and the system is avoided. The method is easy to understand, and the calculation is simple. However, it may also pay excessive attention to some components which have little effect on the reliability, and increase the amount of computation to some extent. Delta-Star and Star-Delta conversion of reliability networks are introduced in [44]. The network equivalence method is also introduced in references [45–50].

(2)    Failure Mode and Effect Analysis

Failure mode and effect analysis (FMEA) is a classical method for reliability assessment [51] that is suitable for radial distribution networks. First, it is necessary to search out the operating states of all components in the system. Then, one must summarize all the fault condition sets and expected accident consequence sets of the system according to the fault state parameters of the components. Finally, one must calculate the reliability index of the system. The principle of this method is simple and the calculation accuracy is high. However, if the system becomes complex and the number of components increases, the system failure states will increase sharply, and the computational complexity will also

increase exponentially. Therefore, it is difficult for it to be directly used for reliability analysis for complex systems. Reference [52] describes the dynamic system behavior based on FMEA and combines the method of fault tree and minimal cut set, and obtains the weak link of the DC system through sensitivity analysis. Some examples of the failure mode and effect analysis method are given in references [53–59].

(3)　Minimal Cut Set Method

The method first obtains the minimal cut set from the power supply to the load. Then, the components in the minimal cut set are assumed to fail one by one, and the reliability index is calculated according to the load loss caused by component faults. This method makes use of the cut set partition and reduces the computation. However, this method can only be applied to simple networks, and it becomes difficult to obtain the minimum cut set for complex networks. In references [60–69], the application of the minimal cut set method is introduced.

(4)　Minimum Path Method

The method first calculates the minimum path from each load point to the power supply, and at the same time converts the influence of the component failure on the non-minimum path to the minimum path corresponding to the load point according to the network topology, and then calculates the reliability index. The minimal path method can better evaluate the reliability of the power system with backup power supply, branch protection, segment switches, and so on. However, this method is only suitable for the reliability evaluation of the system with simple topology. Some applications of the minimum path method are mentioned in references [70–77].

(5)　GO Methodology

The GO methodology was originally proposed for the reliability analysis of weapons and missile systems [78] and was later developed by Matsuoka et al. [79,80] into the GO-flow methodology for the analysis of system reliability. The evaluation process of this method is to first find out the conditions that can guarantee its normal power supply for a certain load or subsystem, and then calculate the probability of realizing the operating conditions as the probability of the normal power supply for the load. This method is easy to understand and simple to apply. However, it may be difficult to calculate the probability of normal operation of various factors such as lines and switches that ensure reliable power supply to the load. In references [81–85], some applications of the GO methodology are introduced.

3.2.2. Simulation Method

Stochastic simulation, also known as Monte Carlo method, was developed in the 1940s. The method is based on the theory of probability and statistics with random sampling as the main method of calculation. It has been applied to power system reliability assessment since 1958 [86,87]. The simulation method is suitable for situations where the system topology is complex and it is difficult to directly establish the reliability model of the system. This method obtains a large number of system reliability estimates through random simulation, and the accuracy of the calculation results is proportional to the number of simulations. The simulation method is generally completed by computer programming, sampling the components of the system, and then calculating the reliability index of the system when the component failure state is randomly generated. This method has obvious advantages when dealing with large-scale systems and has become a main evaluation method [88–90]. However, the calculation is not accurate enough and is time-consuming. A brief flow chart of the simulation method is shown in Figure 4. Simulation methods used in power system reliability assessment include the sequential Monte Carlo method, the non-sequential Monte Carlo method, and the pseudo-sequential Monte Carlo method.

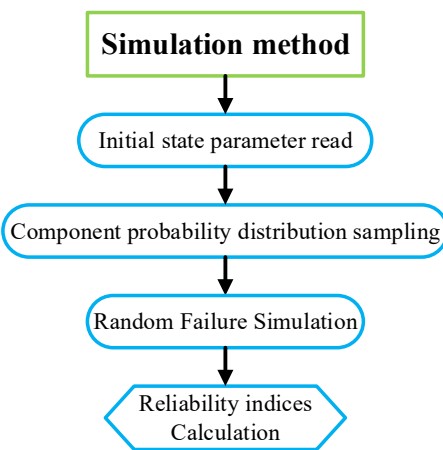

**Figure 4.** Flow chart of simulation method.

(1)  Sequential Monte Carlo Simulation

The sequential characteristics of the power system operation are taken into account by the sequential Monte Carlo simulation method, and the parameter distribution corresponding to the reliability of the system components is sampled based on the time sequence, which better simulates the actual operating conditions and operating characteristics of the system. At the same time, based on the failure and repair parameters of the component, the operation of the component is sampled, and a repair curve is formed. Afterwards, based on the location of the components in the network, the possible effects are determined, and a random sequence of system operating states is obtained. Because this method takes the sequence into account, the model is complicated, and the solution time is also increased. Some reliability evaluation methods based on the sequential Monte Carlo simulation are introduced in references [91–99].

(2)  Non-sequential Monte Carlo Simulation

Non-sequential Monte Carlo simulation is an offline simulation method [100] that does not consider the sequential characteristics of power system operation, but does include random sampling of components in the power system. The non-sequential Monte Carlo simulation method can be divided into the state sampling method and the state transformation sampling method [101]. The state of the components of the system is represented by random numbers drawn from a uniformly distributed array between [0, 1] in this method, and then compared with the probability value of the operating state of each component to determine the operating state of the system. This sampling method has short computational time, fast convergence, and needs low computer memory [102]. The state change sampling method considers the relationship before and after the state change of the system rather than the single state change process of the system components [103]. This method is usually used for the case where the operating state duration of the components follows an exponential distribution, but the calculation time is longer than the previous method. In addition, the reliability evaluation method of pseudo-sequential Monte Carlo simulation is also mentioned in references [104–109].

(3)  Pseudo-sequential Monte Carlo Simulation

The evaluation method is also proposed to improve the shortcomings of the sequential Monte Carlo method and the non-sequential Monte Carlo method, such as slow convergence and large memory usage [103]. Based on the state transition sampling technique of the whole system, the method randomly selects a set of sequences and extracts the random state of the system from this set of state sequences. If the fault state is extracted, the two simulation methods of forward chronological simulation and backward chronological simulation are used to calculate and detect the sub-sequence to which the extracted system fault state belongs. Finally, an actual frequency index of system reliability is calculated. In

references [110–117], some applications of the pseudo-sequential Monte Carlo simulation are introduced.

### 3.2.3. Hybrid Method

Both the simulation method and the analytical method have their advantages and disadvantages and are combined as a hybrid method [118,119]. The idea of the hybrid method is to use the analytical method as much as possible. When the scale of the system is large, the analytical method is difficult to solve; the method requires the use of the simulation method for analysis, which can reduce the calculation variance of the simulation method and shorten the calculation time. The flow chart of the hybrid method is presented in Figure 5. This method takes into account the computational accuracy of the analysis and can also solve large-scale systems and reduce the computational time. However, the research on the reliability assessment of the distribution system by the hybrid method is not deep enough, and further research is needed.

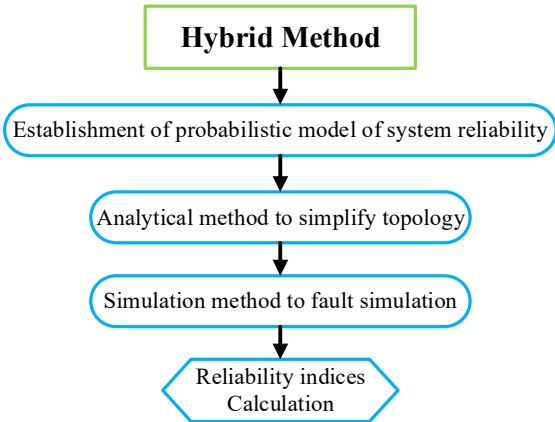

**Figure 5.** Flow chart of the hybrid method.

### 3.3. Natural Gas System Reliability Evaluation

Natural gas plays an important role in people's life and social construction. It is an indispensable energy source for people's daily life and social industry operation. While natural gas brings convenience to people, its security and transport reliability also need to be paid attention to. Although some reliability indices of natural gas systems have been proposed, there is still no authoritative and unified evaluation method. In reference [120], a mathematical model for reliability evaluation of a complex natural gas pipeline network is established by combining stochastic simulation with network flow. In [121], the reliability of the gas supply in the natural gas pipeline network is evaluated by the recursive decomposition algorithm. The staff of Petrobras conducted a reliability analysis of the natural gas pipeline network in Brazil by means of stochastic simulation based on the Pipeline Studio software [122]. In reference [123], researchers introduced a general scientific method to evaluate the reliability of gas supply and oil supply pipe networks, which mainly includes probabilistic mathematical analysis, thermal-hydraulic analysis, and structural integrity analysis, and carried out the pilot application in Kaunas City of Lithuania. In references [124–129], the probabilistic approach and deterministic approach to the assessment of natural gas pipeline reliability were introduced. A method based on the GO methodology to evaluate the reliability of the natural gas transmission station was proposed in [130]. In [131], an approach based on the fault tree analysis method was applied to natural gas system reliability analysis, and its application to liquefied natural gas receiving terminals, which introduced the idea of the reliability of large-scale natural gas pipeline networks, was proposed in [132]. In [133], the prediction of natural gas pipeline system reliability based on the geometry of chaos phase-space restructure and the information geometry support vector machine algorithm was proposed.

*3.4. Integrated Electricity-Gas System Reliability Evaluation*

Domestic and foreign scholars have carried out in-depth research on the reliability of the power system and the natural gas system in recent years. The reliability evaluation methods for the natural gas system can also use the above-mentioned analytical and simulation methods. However, there are only a few research studies on the reliability of IEGS. The IEGS reliability assessment can perform qualitative or quantitative calculations on the integrated energy systems to reflect the risk level of the coupled system. Similar to the reliability assessment of the power system, the security of the energy supply needs to be ensured through the IEGS reliability assessment. If the propagation speed and response speed of energy are ignored, the power supply system and the natural gas system have similarities in energy supply, and the concepts and basic methods of reliability assessment are similar in most aspects. Therefore, most of the reliability assessment theories and indicators of the power grid are used in the reliability assessment of natural gas systems in the existing references [134,135]. Furthermore, complete and unified reliability indicators and evaluation methods have not yet been proposed.

The reliability assessment method of IEGS is mainly divided into two reliability assessment methods: the analytical method and the simulation method. Various methods included in the analytical method and the simulation method in power system reliability evaluation can also be extended to the reliability assessment of IEGS. However, due to the difference of time scale between electric power and natural gas, equipment failures in the network have different effects on the two systems. The focus for IEGS reliability assessment is on the modeling of coupled devices. When modeling a coupled device, in addition to considering the failure of the device itself, it is also necessary to consider the impact of one system failure on the other, that is, the cross-system propagation of the failure.

The concept of an energy hub was proposed for the first time in reference [136] to establish a mathematical model and apply it to the integrated energy system to analyze the reliability of multi-carrier energy systems. Based on the research of scholars, the authors used energy centers to describe the coupling of different energy sources in [137], and modelled the reliability of energy centers according to the process of Markov state transition. The energy flow model including P2G equipment and gas turbine was established in reference [138], and the non-sequential Monte Carlo simulation method was used to evaluate the reliability of the integrated gas-electricity system with wind energy. A hierarchical decoupling optimization framework was proposed in [139], and the influence increment state enumeration method was used to evaluate the reliability of integrated energy systems. This method reduces the proportion of high-order states, and is more accurate and converges faster than the traditional Monte Carlo simulation and state enumeration methods. In [3], the reliability model was established by model-driven and data-driven methods, and the reliability assessment of the integrated energy system was analyzed. In reference [140], a risk assessment model for the IEGS was established with the off-load value of the power system and the natural gas system as the objective function, and some reliability indices were proposed to measure the risks. The similarities and differences in reliability modeling of the power system and the natural gas system were analyzed in reference [141], where a method for reliability evaluation of the IEGS was proposed based on the traditional method for reliability evaluation of the power system.

The operation of a gas-fired combined cycle power plant was studied in [142], and the reliability of the power system considering the influence of the natural gas system was evaluated. In reference [143], a reliability evaluation model for the IEGS was established based on the sequential Monte Carlo simulation method, which optimizes the total cost of the power system and the natural gas system. The criterion of probabilistic reliability was considered in the day-ahead security-constrained unit commitment model of the IEGS in [144]. In reference [145], a reliability evaluation method considering the coupling characteristics of the IEGS was proposed, which adopts the Nataf transformation to deal with the variables that cannot be used directly in the point estimation method. The reliability of the integrated energy system was evaluated based on the Monte Carlo

simulation method in [135], and a reliability model based on smart agent communication was proposed. An integrated energy microgrid model was developed and the reliability of integrated energy systems was evaluated based on the Monte Carlo method in reference [51]. In view of the different energy sources in different subsystems of an integrated energy system, the concept of exergy was proposed to unify the energy sources in different subsystems in reference [146]. Aiming at the problem of failure propagation through the coupling equipment in the power system and the natural gas system, a reliability evaluation method based on the fault of the coupling equipment was proposed in reference [147] to evaluate the reliability of the integrated energy system. A reliability evaluation model for distributed integrated energy systems was developed in reference [148], which was solved by the Monte Carlo method. The probabilistic reliability criterion was added to the robust collaborative planning model of the IEGS in [149] to select the scheme that satisfies the reliability requirements. A reliability-based IEGS planning model was proposed in reference [150] to select the most economical planning decision for transmission lines and natural gas pipelines that satisfies system reliability. A reliability model considering gas turbines and power-to-gas equipment was established in [151], where sequential Monte Carlo simulation was used. An approach based on Monte Carlo simulation techniques for the reliability evaluation of IEGS considering the interdependence-induced cascading effects was proposed in [152]. In [153], the sequential Monte Carlo simulation was used to evaluate the long-term reliability of the IEGS. A short-term reliability evaluation technique for IEGS considering the gas flow dynamics based on the time-sequential Monte Carlo simulation technique was proposed in [154].

Through several decades of development, the indices and methods for power system reliability assessment have matured. Evaluation methods for power system reliability can be divided into simulation methods and analytical methods. However, reliability assessment of natural gas systems has not received the same attention as power systems, resulting in slower development. According to the similarity of the natural gas network and the power system network, some European countries use the Monte Carlo simulation method, the failure mode, and the effect analysis method to evaluate the reliability of the natural gas network. Similarly, these two main reliability assessment methods can also be extended to IEGS. For the convenience of readers, the Table 2 is a summary of the literature on reliability assessment methods for different systems.

**Table 2.** Reliability assessment method reference classification.

| System | Reliability Evaluation Method | | References |
|---|---|---|---|
| Power system | Analytical Method | Network Equivalence Method | [19,44–50] |
| | | Failure Mode and Effect Analysis | [14,52–59] |
| | | Minimal Cut Set Method | [60–69] |
| | | Minimum Path Method | [70–77] |
| | | GO Methodology | [79–85] |
| | Simulation Method | Sequential Monte Carlo Simulation | [91–99] |
| | | Non-sequential Monte Carlo Simulation | [90,101–109] |
| | | Pseudo-sequential Monte Carlo Simulation | [103,110–117] |
| | Hybrid Method | | [118,119] |
| Natural gas system | Analytical Method | | [121,123–125,129–132] |
| | Simulation Method | | [120,122,126,128,133] |
| Integrated electricity-gas system | Analytical Method | | [136,139,141,142,144,145] |
| | Simulation Method | | [135,137,140,143,146,151–154] |

## 4. Conclusions and Outlook

Reliability evaluation is of great significance to the operation and planning of the power system and the natural gas system. It can effectively reduce short-term risks and

ensure the secure and reliable operation of the system. The importance of reliability assessment is first introduced in this paper. Then, existing research studies are reviewed and summarized from the perspective of reliability assessment indicators and reliability assessment methods for power systems and natural gas systems. The establishment of the reliability indices of a single system has been relatively complete, and there are many reliability assessment methods that can guide the daily operation and planning of the system.

The reliability research on electric power has been relatively in-depth. However, due to the large network, complex coupling characteristics, and different energy properties of integrated gas-electric systems, it is still difficult to establish a unified reliability assessment method for integrated gas-electric systems. There are still many difficulties in the establishment of reliability indices, the development of evaluation methods, the reliability modeling of coupled equipment, and the analysis of the influence of fault propagation across systems, which all need to be solved in the future research studies on reliability evaluation of the integrated electricity-gas system.

**Author Contributions:** Conceptualization, C.H.; validation, Y.Z. and C.H.; writing—original draft preparation, Y.Z.; writing—review and editing, Y.Z. and C.H.; supervision, C.H. All authors have read and agreed to the published version of the manuscript.

**Funding:** This work was supported in part by the Science and Technology Project of Sichuan Province 2021YFSY0051.

**Data Availability Statement:** Data available on request from the authors.

**Conflicts of Interest:** The authors declare no conflict of interest.

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
