# Peer review of "A Review on Reliability of Integrated Electricity-Gas System"

_energies, doi:10.3390/en15186815_

Round 1

Reviewer 1 Report

The manuscript lacks certain important features of a good review. Some of them, for example, are discussed below:

1. A tabulated comparison of existing literature highlighting various aspects of the field

2. Couple of generic diagrams so that the readers may mentally map the reviewed articles and can imagine their contribution and field. Similarly, one or two flowcharts are helpful in better understanding of reviewed topic.

3. Some of the other attributes (e.g. a comprehensive outline, timeliness, a conceptual rhythm, etc.) also help making a good review.

So, in my point of view, the manuscript is not in a form which is acceptable for further stages of publication. I suggest a thorough revision of the article. 

Author Response

The authors would like to express our sincere appreciations of the Reviewer’s comments concerning our paper. As the Reviewer concerned, there are several issues that need to be addressed. According to the suggestions of the Reviewer, we have made extensive discussions and corrections to our previous draft.

Reviewer 2 Report

This paper presents the concept of an integrated electricity-gas system and how to evaluate the reliability indices of this system. The authors give almost important reliability indices for power system and natural gas system. Also, reliability evaluation methods are shown, and several articles are reviewed as examples of these methods. However, this article should be improved in terms of content.

  • The author should provide specific data examples of reliability indicators, such as the failure rate of cables under different working conditions or in different countries. This will make the review richer.
  • Reliability evaluation methods should be illustrated with figures describing the basic knowledge of each method. 

Author Response

The authors would like to express our sincere appreciations of the Reviewer’s comments concerning our paper. As the Reviewer concerned, there are several issues that need to be addressed. According to the suggestions of the Reviewer, we have made extensive discussions and corrections to our previous draft

Reviewer 3 Report

I have some comments and suggestions for the paper improvement:

Section 2 - Reliability Indices: Lines 268 to 269, it is said that, according to the NUREG-0492, “the distribution of faults obeys the exponential law”. The document NUREG-0492 states that the failure times follow the exponential law of probabilities under some assumptions; I suggest authors can to include these assumptions.

Section 3 - Reliability Evaluate Method:

a)  Subsections 3.1 and 3.2 have the same title;  

b) Subsection 3.2.1 – Analytical methods: Because this is a review paper, please provide more references regarding the applications of methods 1) Network Equivalence Method, 2) Failure Mode and Effect Analysis, 3) Minimal Cut Set Method, 4) Minimum Path Method.

c)  Subsection 3.2.2 – Simulation method: Because this is a review paper, please provide more references f regarding the applications of methods 1) Sequential Monte Carlo Simulation, 2) Non-sequential Monte Carlo Simulation, 3) Pseudo-sequential Monte Carlo Simulation

d)        Section 3.3 - Natural Gas System Reliability Evaluation: Because this is a review paper, I suggest that authors must to provide more references for the applications of the reliability assessment methods.

1 Abstract – line 13: It is said that paper will include some practical examples about reliability evaluation methods of both systems; I expected to find practical examples of the application of reliability evaluation methods in combined electricity-gas systems, including data and results. Please clarify this point.

Author Response

(The authors gave the same response as above.)

Round 2

Reviewer 1 Report

The authors have significantly improved the manuscript. In my point of view, it has necessary features of a review article. However, addition of generic flow charts in sections 3.2.1, 3.2.2, and 3.2.3 would be an interesting addition.

Author Response

The authors would like to sincerely thank the Editor and the Reviewers’ hard work and helpful comments for improving the quality of this paper. The paper has been carefully revised based on the comments from the Reviewers. In this revised version, changes to the manuscript were all emphasized within the document by using red-colored text. Point-by-point responses to the three Reviewers are listed below.
